# Exploiting model capacity by constraining within-batch features to be orthogonal

**Hyo-Eun Kim**
Lunit Inc.
Seoul, South Korea
hekim@lunit.io

## Abstract

Deep networks have been shown to greatly benefit from large model capacity when trained using various recent deep learning techniques. But at the same time, features in such large capacity networks have a potential to be redundant. In this work, we propose a new regularization method to exploit the given network capacity effectively. By minimizing the redundancy among in-layer filters and the correlation between in-batch features at the same time, we are able to achieve better performance with the same network architecture. Experiments with CIFAR-10/100 show that simultaneously constraining both the in-layer filters to be orthonormal and the in-batch features to be orthogonal is beneficial in efficiently utilizing the model capacity.

## 1 Introduction

Recent trends in deep learning show that deeper model performs better in general (Krizhevsky et al., 2012; Simonyan & Zisserman, 2015; He et al., 2016). This owes to the efforts to resolve vanishing/exploding gradients problem in deep models, such as normalizing outputs of layers to be identical distributions (Ioffe & Szegedy, 2015), connecting layers with shortcuts to make direct paths for gradient flow (He et al., 2016), and initializing weights appropriately to prevent vanishing (or exploding) of gradients (Saxe et al., 2014; Mishkin & Matas, 2016). Although these works enabled learning of deep models, learned filters of deeper networks with large capacity are more likely to be redundant compared to shallow ones. Removing such redundancies would allow us to use the given model capacity effectively, and achieve better generalization with the same models.

(Xie et al., 2017) proposed a method to regularize in-layer filters to be orthonormal during training to solve the gradient vanishing/exploding problem in plain (i.e. without shortcuts) deep networks. The orthonormality among filters in layer-$k$ makes the output features from the layer-$k$ have small correlation with each other and enhances diversity among the filters. However, regularizing the in-layer *filters* to be orthonormal does not guarantee the minimum correlation between the *features*.

In this work, we present a method to exploit the given model capacity effectively by constraining *in-batch features* to be orthogonal and *in-layer filters* to be orthonormal. Since the output features of the layer-$k$ are generated from the input data via all of the preceding layers, this effectively helps to regularize all the relevant filters globally. The proposed method keeps the in-batch features from the layers in a set $S_{feature}$ to be orthogonal and the filters of the layers in a set $S_{filter}$ to be orthonormal at the same time. Experimental results show that the proposed method gives further performance benefit beyond the original orthonormal filter regularizer.

## 2 Methodology

Objective function to guarantee the orthonormality between filters in each layer can be described as,

$$L_{filter} = \frac{1}{N_l} \sum_{l \in S_{filter}} \frac{1}{n_{out}^2} ||W_l W_l^T - \mathbf{I}||_2^2, \qquad W_l \in \mathbb{R}^{n_{out} \times n_{in}}, \qquad (1)$$

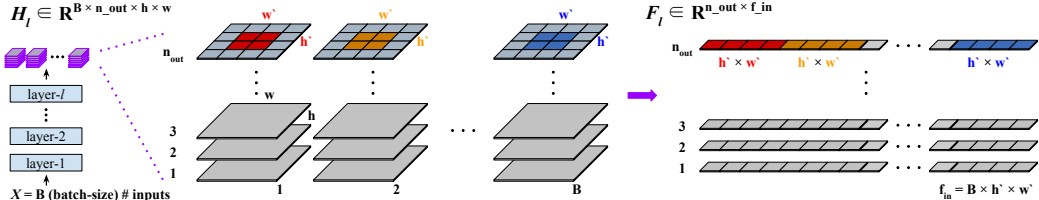

Figure 1: Transformation from the 4-dimensional feature map $H_l$ of layer-$l$ (left) to the 2-dimensional feature matrix $F_l$ (right). Concatenation of center-cropped features across batch-axis defines per-filter representative features.

where $S_{filter}$ is a set of target layers to be optimized via orthonormal filter regularizer, and $N_l$ is the number of layers in the set. $W_l$, $n_{in}$, and $n_{out}$ are a learnable filter bank of $l-th$ layer, the number of parameters in each filter, and the number of filters in $W_l$. For example, $3 \times 3$ convolutional kernel with 64 filters for the 32-channel input feature map; $n_{in}$ and $n_{out}$ are 288 ($= 3 \times 3 \times 32$) and 64 respectively. $\mathbf{I} \in \mathbb{R}^{n_{out} \times n_{out}}$ in Eq. (1) is an identity matrix. During training, correlation between $n_{out}$ number of $n_{in}$ dimensional filters is minimized by the loss function $L_{filter}$.

In order to define the features to be orthogonalized during training, 4-dimensional output feature map $H_l \in \mathbb{R}^{B \times n_{out} \times h \times w}$ of the target layer-$l$ is transformed to the 2-dimensional feature matrix $F_l \in \mathbb{R}^{n_{out} \times f_{in}}$ as shown in Figure 1. $B$ is the batch-size and $f_{in}$ is the dimension of each transformed feature vector (i.e. each row of $F_l$). Since each transformed feature vector should be representative of each corresponding filter, we define the per-filter feature as a concatenation of the features of all the input examples in a batch. Each per-example feature is center-cropped as shown in Figure 1. Target loss for the orthogonality among features is as follows,

$$L_{feature} = \frac{1}{N_l} \sum_{l \in S_{feature}} \frac{1}{n_{out}^2} ||\hat{F}_l \hat{F}_l^T - \mathbf{I}||_2^2, \quad \hat{F}_l \in \mathbb{R}^{n_{out} \times f_{in}}, \tag{2}$$

where $S_{feature}$ and $N_l$ are a set of target layers for the in-batch feature orthogonalization and the number of layers in the set, respectively. $F_l$ is the target in-batch feature transformed from the output feature $H_l$, and $\hat{F}_l$ is the unit-vectorized version of $F_l$ (i.e. each row of $\hat{F}_l$ is a unit vector converted from corresponding row of $F_l$). $\hat{F}_l \hat{F}_l^T$ is a matrix with ones on a diagonal, thus the orthogonality between features of $F_l$ is guaranteed by this loss.

Incorporating both $L_{filter}$ and $L_{feature}$ with a task solving loss $L_{task}$ (e.g., negative-log-likelihood for classification), total loss is described as,

$$L = (1 - \lambda_1 - \lambda_2)L_{task} + \lambda_1 L_{filter} + \lambda_2 L_{feature}, \tag{3}$$

where the loss weighting constants $\lambda_1$, $\lambda_2$, and their sum $\lambda_1 + \lambda_2$ should be in the range of [0,1].

## 3 EXPERIMENTS

We conducted experiments on the CIFAR-10/100 datasets, which have 10/100 classes with 50k/10k training/test images, respectively (Krizhevsky & Hinton, 2009). First, we experimentally confirmed that the orthonormal filter regularization is effective not only in the plain deep models but also in deep models with residual connections.[1] Based on our experiments, ResNet with both L2 and orthonormal filter regularizers is better than the model with only the orthonormal filter regularizer. For example, error rate with mean (std) of 4 trials in cifar-10; ResNet-32 with orthonormal filter

---

[1](Xie et al., 2017) also showed the effectiveness of the orthonormal filter regularization in deep networks with shortcuts; e.g., ResNet-110 with the orthonormal filter regularizer (error rate: 10.0%) is better than original ResNet-110 with L2 regularizer (13.5%), but the error rate reported in the original paper (He et al., 2016) was much smaller (6.61%).

Table 1: Classification error rates (%) of 4 trials with mean (std) on CIFAR-10/100; *Baseline* refers to the original ResNet (He et al., 2016), *Filter* refers to the orthonormal filter regularizer (only) (Xie et al., 2017), *Feature* refers to the orthogonal feature regularizer (only), and *Both* refers to both of the regularizers.

| | cifar-10 | | | | cifar-100 | | | |
| --- | --- | --- | --- | --- | --- | --- | --- | --- |
| | ResNet-32 | ResNet-56 | ResNet-110 | ResNet-152 | ResNet-32 | ResNet-56 | ResNet-110 | ResNet-152 |
| Baseline | 7.57 (0.207) | 6.98 (0.306) | 6.60 (0.340) | 6.44 (0.349) | 31.43 (0.211) | 28.15 (0.412) | 26.82 (0.531) | 26.71 (0.683) |
| Filter | 7.64 (0.128) | 6.64 (0.158) | 6.00 (0.097) | 5.85 (0.220) | 31.12 (0.519) | **27.46 (0.272)** | 26.08 (0.433) | 25.50 (0.166) |
| Feature | 7.85 (0.128) | 6.69 (0.129) | 6.20 (0.135) | 6.41 (0.265) | 31.46 (0.224) | 27.85 (0.127) | 26.16 (0.161) | 25.76 (0.157) |
| Both | **7.44 (0.131)** | **6.34 (0.132)** | **5.78 (0.173)** | **5.84 (0.048)** | **31.00 (0.365)** | 27.61 (0.240) | **25.63 (0.181)** | **25.13 (0.270)** |

regularizer only: 8.20% (0.246), ResNet-32 with both orthonormal filter and L2 regularizers: 7.64% (0.103), so we use both regularizers for reproducing (Xie et al., 2017) in our experiments.

(He et al., 2016) introduced two types of residual blocks; basic block (consists of 2 consecutive convolutions), bottleneck block (consists of 3 consecutive convolutions with bottleneck structure). Bottleneck-type ResNet performs similar to the basic-type with much smaller parameters in ResNet-56, 110, 152, and the basic-type performs better in ResNet-32; e.g., bottleneck-type ResNet-56 (# params: 0.59M, error rate: 6.98%), basic-type ResNet-56 (# params: 0.86M, error rate: 6.99%). For our baseline architecture, bottleneck-type is used for ResNet-56, 110, 152 and basic-type is used for ResNet-32. Details of the architecture comparison are summarized in Appendix (Table 2 and 3).

Augmentation follows the original paper (He et al., 2016); images are zero-padded with 4 pixels on each side, randomly cropped to produce 32×32 images, flipped horizontally with probability 0.5, and normalized by subtracting channel means and dividing by channel standard deviations.

$S_{filter}$ includes all the convolution layers and $S_{feature}$ includes all the convolution layers except for the first convolution layer, and $w'$ and $h'$ in Figure 1 are 2. Learning rate 0.1 is decayed by $\frac{1}{10}$ at $80-th$ and $120-th$ epoch during training until $160-th$ epoch. The weight decay constant is 0.0001 and stochastic gradient descent with momentum 0.9 is used. Based on 50k training images, performance on the 10k test images at the last ($160-th$) epoch was reported. All experiments are conducted with PyTorch.[2]

Error rates of 4 trials with mean (std) are summarized in Table 1 (details of each model are described in the caption). $\lambda_{1,2}$ in Eq. (3) of *Baseline*, *Filter*, *Feature*, and *Both* are (0.0, 0.0), (0.25, 0.0), (0.0, 0.25), and (0.25, 0.25), respectively. *Feature* is beneficial, but worse than *Filter* in general. And *Both* performs the best. The result of *Both* implicitly shows that the source of the performance benefit in *Feature* is different from that of *Filter*. We investigated the average correlation between features with respect to each model (details are in Appendix; Table 4). In summary, orthonormality among filters helps to reduce the correlation between features but the gap between *Baseline* and *Filter* is small. The average correlation of *Both* is the lowest among the models, which is slightly lower than *Feature*; i.e. both *Filter* and *Feature* contribute complementarilly to achieve the best performance.

## 4 CONCLUSION AND FUTURE WORKS

We showed that constraining the in-batch features to be orthogonal and the in-layer filters to be orthonormal can help to exploit given model capacity effectively. Directly handling the output features to be orthogonal is beneficial, and the best performance is achieved when the orthonormal filter regualizer is added. Although the proposed method is shown to be effective with the small datasets, further experimental validation with large-scale datasets is needed. Our future work includes more experimental analysis with different configurations (e.g., using convolution layers in the last residual block for $S_{feature}$ shows better performance in ResNet-152; 5.72 (0.178)) and datasets (e.g., ImageNet).

---

[2]https://github.com/pytorch/vision/tree/master/torchvision/models

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

## 5 APPENDIX

Table 2 compares two residual blocks in ResNet (He et al., 2016), and the performance with the model capacity of the two types are in Table 3. Table 4 shows average correlation between features.

Table 2: ResNet architecture for CIFAR-10/100; basic block vs bottleneck block. Downsampling with stride 2 is performed by res-block2 and res-block3.

| layer name | output size | basic block | bottleneck block | # blocks |
|---|---|---|---|---|
| init conv | $32 \times 32$ | $3 \times 3$, 16 (stride=2) | | |
| res-block 1 | $32 \times 32$ | $\begin{bmatrix} 3 \times 3, 16 \\ 3 \times 3, 16 \end{bmatrix}$ | $\begin{bmatrix} 1 \times 1, 16 \\ 3 \times 3, 16 \\ 1 \times 1, 64 \end{bmatrix}$ | $\times n_1$ |
| res-block 2 | $16 \times 16$ | $\begin{bmatrix} 3 \times 3, 32 \\ 3 \times 3, 32 \end{bmatrix}$ | $\begin{bmatrix} 1 \times 1, 32 \\ 3 \times 3, 32 \\ 1 \times 1, 128 \end{bmatrix}$ | $\times n_2$ |
| res-block 3 | $8 \times 8$ | $\begin{bmatrix} 3 \times 3, 64 \\ 3 \times 3, 64 \end{bmatrix}$ | $\begin{bmatrix} 1 \times 1, 64 \\ 3 \times 3, 64 \\ 1 \times 1, 256 \end{bmatrix}$ | $\times n_3$ |
| final block | $1 \times 1$ | avgpool, cls#-dim fc, softmax | | |

Table 3: Comparison of the two types; # of blocks, # of paramerters, and error rate (%) with mean (std) of 4 trials. Bottleneck-type ResNet performs similar to the basic-type ResNet with much smaller parameters.

| | | ResNet-32 | ResNet-56 | ResNet-110 | ResNet-152 |
|---|---|---|---|---|---|
| basic block | $(n_1, n_2, n_3)$ | (5, 5, 5) | (9, 9, 9) | (18, 18, 18) | (25, 25, 25) |
| | # params | 0.47M | 0.86M | 1.73M | 2.41M |
| | error rate (%) | 7.57 (0.207) | 6.99 (0.100) | 6.54 (0.203) | 6.40 (0.113) |
| bottleneck block | $(n_1, n_2, n_3)$ | (3, 3, 4) | (6, 6, 6) | (12, 12, 12) | (16, 16, 18) |
| | # params | 0.38M | 0.59M | 1.15M | 1.66M |
| | error rate (%) | 7.80 (0.287) | 6.98 (0.306) | 6.60 (0.340) | 6.44 (0.349) |

Table 4: Average correlation between features; mean of $L_{feature}$ at the last epoch (4 trials). Orthonormality between filters in *Filter* helps to reduce the average correlation of features from *Baseline*, but the gap is small. *Both* shows the lowest correlation which is slightly lower than *Feature*.

| | cifar-10 | | | | cifar-100 | | | |
|---|---|---|---|---|---|---|---|---|
| | ResNet-32 | ResNet-56 | ResNet-110 | ResNet-152 | ResNet-32 | ResNet-56 | ResNet-110 | ResNet-152 |
| Baseline | 0.121 | 0.103 | 0.143 | 0.147 | 0.101 | 0.089 | 0.113 | 0.136 |
| Filter | 0.114 | 0.096 | 0.110 | 0.118 | 0.091 | 0.084 | 0.099 | 0.111 |
| Feature | 0.034 | 0.039 | 0.040 | 0.041 | 0.026 | 0.040 | 0.042 | 0.045 |
| Both | 0.021 | 0.032 | 0.033 | 0.034 | 0.021 | 0.034 | 0.035 | 0.037 |

