# OpenReview forum: "Exploiting model capacity by constraining within-batch features to be orthogonal"
_ICLR.cc/2018/Workshop — Reject_

### Official Review · AnonReviewer1 · 2018-03-07
**Very Incremental**

**Rating:** 5
**Confidence:** 4

**Review:**

Authors in this paper builds on top of the previous orthogonal filter regularization and add a similar regularization on the feature maps of the CNN. Experiments on CIFAR10 and CIFAR100 show that the proposed technique can slightly improve the generalization.

I have a few concerns:
(1) Overall, the contribution of this paper is very incremental and experimental results are not significant. Given that the technical contribution is weak, I recommend you add more experiments to make the paper more convincing, e.g., adding comparison on ImageNet.
(2) From your regularization in Eq. (1), it only ensures the filter to be orthogonal rather than orthonormal, i.e., filters are not unit vectors. The naming is problematic.

---

> ### Public Comment · ~Hyo-Eun_Kim1 · 2018-03-20
> **Answer to the Reviewer1**
>
> Eq.(1): W W^T = I ensures that the original filters in W to be "orthonormal" (not orthogonal). It is clear.
>
> \hat(W) \hat(W)^T = I (your recommendation; \hat(W) = unit-vectorized W) ensures that the original filters in W to be "orthogonal", because it indirectly regularizes the original filter vectors by regularizing the unit-vectors of W to be orthonormal.
>
> Thanks for your comment.

---

> > ### Comment · AnonReviewer1 · 2018-03-26
> > **Reply to Your Answer**
> >
> > Sorry for my mistake on the unit norm. I am actually about to ask the following question. Since the filter W is typically not square, what is the difference between regularize | WW^T - I |^2 and | W^TW - I |^2?

---

> > > ### Public Comment · ~Hyo-Eun_Kim1 · 2018-03-26
> > > **Answer to the second question**
> > >
> > > W \in R^{n_out, n_in}, where n_out and n_in are the number of filters (or features) and the dimension of each filter (or feature). So, WW^T \in R^{n_out, n_out} consists of possible correlations between any two filters (or features) among all the n_out number of filters (or features). As you mentioned that W is typically not square, W^T W is not possible in general.
> > >
> > > Now, I'm doing experiments on the imagenet dataset, and the proposed method shows similar (or even better) results compared to the experiments with CIFAR datasets (in the original manuscript). I'll share the results in the near future via arXiv. Thanks!

---

> > > > ### Comment · AnonReviewer1 · 2018-03-26
> > > > **Reply to Your Answer**
> > > >
> > > > Why is W^T W not possible? It will be of shape n_in by n_in, right? It also captures the orthogonality constraint but in your input dimension. I am curious how it differs to your regularization.

---

> > > > > ### Public Comment · ~Hyo-Eun_Kim1 · 2018-03-26
> > > > > **Answer to your comment**
> > > > >
> > > > > Sorry for my mistake. It is possible. The purpose of the proposed method is to reduce the correlation between n_out number of n_in dimensional feature vectors. Each n_in dimensional feature vector is representative of each filter.
> > > > >
> > > > > W W^T \in R^{n_out, n_out} regularizes the filter-representative features to have minimum correlation each other, while reducing correlation between different feature extraction paths. E.g., for the case of z = f3(f2(f1(x))), where f1,2,3 are the 1st, 2nd, 3rd layers, x is an input example, and z is the output feature maps, z_i is the i^th feature map which is representative of the i^th filter of f3. By minimizing the correlation between z_i and z_j (except for the i=j), we can reduce global correlation between different feature extraction paths (from x to f3).
> > > > >
> > > > > I think regularizing W^T W is not appropriate for this purpose, because W^T W \in R^{n_in, n_in} will reduce the correlation between n_out dimensional feature vectors.

---

### Official Review · AnonReviewer3 · 2018-03-10
**This paper is easy to follow, but the experiments lack some important things.**

**Rating:** 6
**Confidence:** 5

**Review:**

This paper seeks to enforce orthonormality over model parameters of deep neural networks.  This is done by introducing two loss terms that reflecting orthonormality over parameters and features, respectively. This paper is easy to follow, but the experiments lack some important things.

The experiments lack results on large data set.

How to set lambda_1 and lambda_2?

I think both lambda_1 and lambda_2 should be in [0,1]

---

### Official Review · AnonReviewer2 · 2018-03-12
**Interesting but very limited work**

**Rating:** 5
**Confidence:** 3

**Review:**

The authors propose  (approximate) orthonormalization of the in-layer features combined with (approximate) orthogonalization of the in-batch features.

They claim (based on evidence on CIFAR-10 and CIFAR-100) that this combination is an effective regularizer.

There is a significant literature of constraining weight matrices to be orthonormal (as it is known to be a good pre-conditioner), but people mostly try hard constraints instead of soft constraints.

There are two drawbacks of the proposed method:
- It is relatively costly to batch-orthogonalize the activations.
- Current best results are achieved with wider architectures (Wide Residual Networks by Zagoruyko and Komodakis) and there is current theoretical evidence that over-parametrization helps optimization.  (On the Optimization of Deep Networks:
Implicit Acceleration by Overparameterization by Arora et al.)

Given the fact that the batch-orthogonalization is quite costly, questionably efficient on wide networks, the baselines on cifar-10 and 100 are not up-to-date (Wide Residual Networks came out in 2016 and reach 3.8% error on cifar-10 and about 20% error on cifar-100), I wonder why this work was not based on that work. For this reason, I am not convinced about the scope of this work.

---

### Decision · Program_Chairs · 2018-03-20
**ICLR 2018 Workshop Acceptance Decision**

**Decision:**

Reject

**Comment:**

Based on the reviews, this paper has not been accepted for presentation at the ICLR workshop. However, the conversation and updates can continue to appear here on OpenReview.